Psychological, pharmacological, and combined treatments for binge eating disorder: a systematic review and meta-analysis

Ghaderi Ata 1 ata.ghaderi@ki.se
Odeberg Jenny 2
Gustafsson Sanna 3 4
http://orcid.org/0000-0003-0819-3946 Råstam Maria 5 6
Brolund Agneta 2
Pettersson Agneta 2
http://orcid.org/0000-0002-6159-598X Parling Thomas 4 7
1 Division of Psychology, Department of Clinical Neuroscience, Karolinska Institute , Sweden
2 Swedish Agency for Health Technology Assessment, and Assessment of Social Services , Stockholm , Sweden
3 University Health Care Research Centre, Faculty of Medical Sciences, University College of Örebro , Örebro , Sweden
4 Centre for Psychiatry Research, Department of Clinical Neuroscience, Karolinska Institute , Stockholm , Sweden
5 Department of Clinical Sciences, Lund University , Lund , Sweden
6 Gillberg Neuropsychiatry Centre, University of Gothenburg , Gothenburg , Sweden
7 Stockholm Health Care Services, Stockholm County Council , Stockholm , Sweden
Zohar Ada
Electronic publication date: 2018 Jun 21
Publication date: 2018
Volume: 6
Electronic Location ID: e5113
Received 2018 Mar 2; Accepted 2018 Jun 6
Copyright: © 2018 Ghaderi et al.
Copyright year: 2018
Copyright holder: Ghaderi et al.
License: This is an open access article distributed under the terms of the Creative Commons Attribution License, which permits unrestricted use, distribution, reproduction and adaptation in any medium and for any purpose provided that it is properly attributed. For attribution, the original author(s), title, publication source (PeerJ) and either DOI or URL of the article must be cited.
License URL: https://creativecommons.org/licenses/by/4.0/

Keywords: Eating disorder, Psychotherapy, Meta-analysis, Binge eating disorder, SSRI, Lisdexamfetamine

Funding: Swedish Agency for Health Technology Assessment, and Assessment of Social Services This work was supported by the Swedish Agency for Health Technology Assessment, and Assessment of Social Services, but we did not receive any specific grant from funding agencies in the public, commercial, or not-for-profit sectors. The funders had no role in study design, data collection and analysis, decision to publish, or preparation of the manuscript.

==============================
Objective

To systematically review the efficacy of psychological, pharmacological, and combined treatments for binge eating disorder (BED).

Method

Systematic search and meta-analysis.

Results

We found 45 unique studies with low/medium risk of bias, and moderate support for the efficacy of cognitive behavior therapy (CBT) and CBT guided self-help (with moderate quality of evidence), and modest support for interpersonal psychotherapy (IPT), selective serotonin reuptake inhibitors (SSRI), and lisdexamfetamine (with low quality of evidence) in the treatment of adults with BED in terms of cessation of or reduction in the frequency of binge eating. The results on weight loss were disappointing. Only lisdexamfetamine showed a very modest effect on weight loss (low quality of evidence). While there is limited support for the long-term effect of psychological treatments, we have currently no data to ascertain the long-term effect of drug treatments. Some undesired side effects are more common in drug treatment compared to placebo, while the side effects of psychological treatments are unknown. Direct comparisons between pharmaceutical and psychological treatments are lacking as well as data to generalize these results to adolescents.

Conclusion

We found moderate support for the efficacy of CBT and guided self-help for the treatment of BED. However, IPT, SSRI, and lisdexamfetamine received only modest support in terms of cessation of or reduction in the frequency of binge eating. The lack of long-term follow-ups is alarming, especially with regard to medication. Long-term follow-ups, standardized assessments including measures of quality of life, and the study of underrepresented populations should be a priority for future research.

Introduction

Binge eating disorder (BED) became a formal diagnosis within the eating and feeding disorders in the fifth edition of the diagnostic and statistical manual for mental disorders (DSM-5: American Psychiatric Association, 2013). BED is characterized by episodes of binge eating defined as eating, in a discrete period of time, an amount of food that is definitely larger than most people would eat during similar circumstances while experiencing a lack of control over eating.

Epidemiological studies suggest that 1–4% in the population suffer from BED (Hoek, 2006; Kessler et al., 2013). The onset is usually in the late adolescent or early adult years (Nicholls, Lynn & Viner, 2011) with an increasing incidence in the late adolescence and early adulthood in both sexes (Hudson et al., 2007; Stice, Marti & Rohde, 2013). Recent studies suggest that the lifetime prevalence of BED (1.4%) is higher than that of bulimia nervosa (0.8%) (Kessler et al., 2013).

Patients with BED often present with comorbid psychiatric and somatic diagnoses, lower quality of life, more suicide ideation and attempts, and lower social functioning compared to the general population (Agh et al., 2015; Favaro & Santonastaso, 1997; Grilo, White & Masheb, 2009; Hudson et al., 2007; Kessler et al., 2013; Ulfvebrand et al., 2015; Wilfley, Wilson & Agras, 2003). In studies using semi-structured interviews, the lifetime prevalence of obesity in BED-patients is 42–49% (Hudson et al., 2007; Kessler et al., 2013), which translates into an increased risk of conditions as diabetes and metabolic syndrome (Citrome, 2017), with increased health care consumption and health care costs (Agh et al., 2015).

In terms of treatment interventions, cognitive behavior therapy (CBT), CBT guided self-help (CBT-gsh), interpersonal psychotherapy (IPT), behavior weight loss (BWL), dialectical behavior therapy (DBT), pharmaceutical treatments and different combinations of medical and psychological treatments have been studied (Fairburn et al., 2015; Grilo, 2017; McElroy, 2017). CBT and CBT-gsh are the most well studied treatments, with a large number of studies. BWL programs have been evaluated in individual (Wilson, 2011), group (Munsch et al., 2007), and self-help program formats (Grilo & Masheb, 2005). Since a majority of BED patients also suffer from overweight or obesity many treatments have not only addressed the binge-eating behavior, but also aimed at weight loss (Agras et al., 1994; Grilo, 2016). Some studies have also investigated the effect of bariatric surgery on binge eating behaviors, in addition to its weight loss effects (Colles, Dixon & O’Brien, 2008; De Man Lapidoth, Ghaderi & Norring, 2011; De Zwaan et al., 2010; McElroy et al., 2011a, 2011b).

In terms of pharmacological treatments, the efficacy of a broad spectrum of pharmacotherapeutic agents on binge eating frequency and weight loss have been investigated (McElroy, 2017). Given the short-term efficacy of anti-depressants, specifically fluoxetine, for bulimia nervosa, anti-depressants for BED has been studied in several randomized controlled trials (RCT). Due to high comorbidity of overweight and obesity in BED (De Man Lapidoth, Ghaderi & Norring, 2006; De Zwaan, 2001; Decaluwe & Braet, 2003; Vamado et al., 1997), the efficacy of anti-obesity agents such as fenfluramine, orlistat, and sibutramine have also been investigated. On the basis of the hypothesized biological underpinnings of BED or similarity with some other conditions such as addiction disorders (McElroy, 2017), other drugs such as antiepileptics (e.g., topiramate and lamotrigine) as well as drugs that are usually prescribed for attention deficit and hyperactivity disorders (e.g., lisdexamfetamine), or anti-addiction drugs (e.g., naloxone) have also been tested with binge eating frequency and weight loss as main outcome variables. Most of the pharmacological trials are short-term interventions (6–16 weeks) with effects investigated at post-treatment, but almost no long-term follow-ups exist.

Several previous reviews and meta-analyses of the treatment of BED have focused on RCTs (Amianto et al., 2015; Brownley et al., 2015, 2016; Grilo, 2017; Iacovino et al., 2012; McElroy, 2017; McElroy et al., 2015b; Reas & Grilo, 2014, 2015; Vocks et al., 2010). Nine RCTs (Brownley et al., 2016) investigated psychological treatments vs. waitlist controls and 25 RCTs investigating the effect of drugs. Significantly more participants achieved abstinence from binge eating with CBT vs. waitlist. Other forms of psychological treatments such as dialectic behavior therapy and behavioral weight loss (BWL) also reduced binge-eating and related psychopathology. CBT (whether delivered in therapist-led, partially therapist-led, or CBT-gsh) did not significantly reduce weight or symptoms of depression. Interestingly, the authors defined many forms of therapy such as psychodynamic psychotherapy as a BED-focused CBT. Further, the review reported that second-generation antidepressants such as citalopram, fluoxetine, and sertraline decreased binge eating and reduced symptoms of depression. Similar outcomes were found for topiramate as well. Lisdexamfetamine reduced binge-eating and binge-eating-related obsessions and compulsions. Only lisdexamfetamine and topiramate reduced weight. However, the authors note the inconsistencies of outcome measures across trials and the paucity of assessments beyond the end of treatment (Brownley et al., 2015). A review evaluating controlled studies of pharmacotherapy for BED found 22 RCTs (Reas & Grilo, 2015) of which 14 were pharmacotherapy-only trials and eight trials investigating combinations with CBT and/or BWL. All but two studies had been reported in an earlier report by the authors (Reas & Grilo, 2014). Reviews of RCTs suggest that the outcome of psychological treatments, alone or in combination with drugs, are superior to drugs only, while the combined treatments of BED (Grilo, 2017; Reas & Grilo, 2014) failed to show superiority compared to CBT only. Nevertheless, adding some drugs to psychological treatments might enhance the level of weight loss, compared to CBT or BWL treatment only, although the effects are modest (Reas & Grilo, 2014). Summing up, reviews of treatment research of BED showed that CBT, individually or in groups, was most researched and produced the best results in terms of proportion of patients reaching remission compared to waiting list. The evidence-base for pharmacotherapy was limited. In the short term SGAs seem to reduce the number of binge episodes, but there are almost no RCTs on its longer-term efficacy.

Aims of the study

Given the importance of replications, the aim of the present study was to evaluate the efficacy and quality of evidence, as well as potential iatrogenic effects of treatment in controlled psychological, pharmacological, and combined treatment interventions for BED. Outcomes of interest were remission, episodes of binge eating, weight loss, measures of specific psychopathology of eating disorders, depressive symptoms, quality of life, and side effects. We included only studies characterized by low or moderate risk of bias.

Methods

The systematic review was conducted in accordance with the PRISMA statement (Moher et al., 2009). The inclusion criteria were as follows: RCT and prospective controlled clinical trials, participants with full or threshold BED according to DSM-IV (American Psychiatric Association, 2000) research criteria, or BED according to DSM-5 (American Psychiatric Association, 2013) regardless of age or weight status, and all types of interventions except for outdated treatments (i.e., studies investigating drugs that have been withdrawn from the market due to serious side effects).

Outcomes of interest were remission, episodes of binge eating, weight loss, measures of specific psychopathology of eating disorders, depressive symptoms, quality of life, and side effects. We did not decide upon imposing any a priori specific inclusion or exclusion criteria regarding the duration of treatment, length of follow-up(s) or number of participants.

Search strategy

The databases PubMed (NLM), Embase (Elsevier), the Cochrane Library (Wiley), Scopus, and the HTA databases from Centre for Reviews and Disseminations were searched until November 2015. The search was updated again in November 2016. Reference lists and books were also used to identify further studies. Search strategies are listed in Appendix A. Two reviewers screened the titles and abstracts independently. Full text articles were retrieved if one or both reviewers considered a study potentially eligible. Both reviewers read the full texts, and consensus was reached regarding eligibility. The excluded articles are provided in Appendix B.

A total of two pairs of reviewers assessed eligible studies for risk of bias independently. Studies were scored as having either high or acceptable (i.e., low or medium) risk of bias. Assessment of the risk of bias of each study was based on the quality of the randomization procedure and equality of the conditions before the treatment, allocation concealment, blinding (participants, assessors, treatment providers), drop-out (<30% of total sample and <10% difference between the conditions), potential conflict of interest, and analysis of confounders. Studies were excluded if they were not controlled, or if the drop-out rate was larger than those reported above. In addition to independent ratings by at least two experts for each paper, whenever any minor issues related to randomization, allocation, blinding, etc. were unclear, they were thoroughly discussed by the entire group to reach consensus on the level of risk of bias. Only studies with acceptable risk of bias were included in this review.

Data management

For all studies, we extracted country, type of setting, method of recruitment, type of treatment, treatment length, number of sessions, treatment format, age, body mass index (BMI), and gender. Extraction from drug treatments also included dosage during the trial. Study characteristics are reported in Appendix C. Outcomes included remission defined as complete cessation of binge eating, BE frequency (i.e., number of binges/week), BMI (weight reduction), specific psychopathology of eating disorders measured by specific measures such as the eating disorder examination questionnaire (Fairburn, 2008), depressive symptoms, and quality of life, and side effects.

Statistical analysis

For dichotomous outcomes, we estimated the risk difference (RD) or risk ratio (RR), and for continuous outcomes we estimated the mean difference (MD) or standard mean difference (SMD). All outcomes were reported with 95% confidence intervals (CI). Dichotomous effects were weighted using the Mantel-Haenszel method and continuous effects were weighted using Inverse Variance.

In one case we contacted the authors and received supplementary data from one study (Schlup et al., 2009), which enabled inclusion in the meta-analysis. Data synthesis was carried out using Review Manager (RevMan) version 5.3 (2014) employing the random effects model due to clinical heterogeneity.

The certainty of evidence was assessed with grading of recommendations assessment, development, and evaluation (GRADE: strong, moderate, low or insufficient) (Guyatt et al., 2008). In brief, preliminary certainty of the evidence was classified as high (labeled ⊕⊕⊕⊕) if the results were based on data from RCT, otherwise the preliminary certainty of evidence starts with low (labeled ⊕⊕ΟΟ). Thereafter, we analyzed to what extent the results from the meta-analysis might be affected by the five risk domains in GRADE. These are: overall risk of bias across studies, degree of heterogeneity between studies (inconsistency), size of the CI for the summary measures (imprecision), risk for publication bias and risk that the results are not generalizable to the actual context (indirectness). The final certainty of the evidence depended on whether there were severe deficiencies in any of the five risk domains. Thus, the resulting certainty of evidence could be high (⊕⊕⊕⊕), moderate (⊕⊕⊕Ο), low (⊕⊕ΟΟ) or insufficient/very low (⊕ΟΟ). One special rule was applied; if an intervention was evaluated in only one small study (size < 100) the certainty of the evidence was a priori assessed as very low/insufficient (⊕ΟΟΟ).

Results

The search of the databases resulted in 3,595 publications and the abstracts of these were screened (See Fig. 1). Of these, 296 were obtained in full text and screened. A total of 99 were then classified as relevant according to the inclusion and exclusion criteria. At this stage we classified the trials according to drug, psychotherapy, or a combination of drug and psychotherapy (1:st classification). Among the selected 99 publications, we found 45 unique studies (54 publications) with low/medium (i.e., acceptable) bias risk. A total of 45 publications were assessed as high bias risk and were not included in the analysis. See Appendix B for excluded publications.

Figure 1 Search results and inclusion of the studies.

Taken together, the most common reasons for exclusion due to high risk of bias were unclear randomization, allocation, whether assessors were blinded and high drop out.

Study characteristics

All of the included studies were RCT’s. The studies included 4,611 participants (range 18–773) and they were conducted in North America and Europe with the exception of one study from South America. Participants were 18 years and older, recruited mainly from out-patient settings and through direct advertisement to the community, and the majority were women. Standardized diagnostic interviews were used to establish BED diagnosis. The majority of studies had a lower inclusion criteria regarding BMI in the overweight to obese weight spectrum. However, the outcome of the treatment of BED was not presented separately for participants with or without concurrent obesity in any of the studies. Please see Appendix C for detailed characteristics of included studies.

Interventions

The trials included a variety of interventions and various combinations of them (drugs, psychotherapy, BWL, and low-energy-density diet counseling) in different formats (individual or group, as well as self-help with or without support) and different control conditions (wait list or placebo). We first classified the trials according to drug, psychotherapy, or a combination of drug and psychotherapy (1:st classification). In the second step, we identified and matched those with similar treatment and control condition. This resulted in the following potential meta-analyses: Drugs vs. placebo (selective serotonin reuptake inhibitors (SSRI), lisdexamfetamine, mood stabilizer, anorexiants), combination of drug and CBT/CBT-gsh vs. placebo, CBT vs. wait list, CBT-gsh vs. wait list, IPT vs. CBT/CBT-gsh, and BWL vs. CBT/CBT-gsh.

Included studies not in the meta-analyses per 1:st classification

Drug vs. placebo: For some of the remaining studies with acceptable risk of bias, only one study per drug was identified. There was insufficient number of studies to assess the efficacy of the following drugs (n < 100): bupropion (Norepinephrine–dopamine reuptake inhibitor, NDRI) (White & Grilo, 2013), dulextin (Serotonin–norepinephrine reuptake inhibitor, SNRI) (Guerdjikova et al., 2012), baclofen (muscle relaxant) (Corwin et al., 2012), imipramine (tricyclic antidepressant) (Laederach-Hofmann et al., 1999), and armodafinil (sympathomimetic) (McElroy et al., 2015a). One study investigated fluoxetine vs. sertraline (Leombruni et al., 2008). Thus, a total of 14 studies were included in the meta-analyses for drugs vs. placebo for BED.

Combination of drug and psychological treatment: The remaining five studies with acceptable risk of bias investigated unique combinations of drug and psychological treatment. All studies were too small (n < 100) and thus there was low certainty of the evidence to assess the following combinations; desipramine (tricyclic anti-depressive) + CBT (Agras et al., 1994), topiramate (anticonvulsive) combined with CBT (Claudino et al., 2007), orlistat (anorexiant) combined with CBT-gsh (Grilo, Masheb & Salant, 2005), orlistat (anorexiant) combined with BWL (Grilo & White, 2013), and fluoxetine (SSRI) combined with CBT (Grilo, Masheb & Wilson, 2005).

Psychological treatment: Of the remaining 24 studies (i.e., 31 publications) with acceptable risk of bias, 13 were not included in any meta-analysis. Alfonsson, Parling & Ghaderi (2015) investigated behavioral activation vs. wait list, and Shapiro et al. (2007) investigated CBT for healthy eating and weight control in group format as one of three arms (the wait list vs. CBT-gsh comparison were included in the CBT-gsh vs. wait list meta-analyses). These two interventions were assessed as providing a too general CBT approach not including the specific eating disorder interventions as the included publications in the meta-analysis for CBT vs. wait list. The mindfulness based eating awareness training condition and psychoeducation with elements of CBT condition vs. wait list (Kristeller, Wolever & Sheets, 2014) were excluded due to too low similarity with the included interventions in the meta-analysis for CBT vs. wait list. In addition, in some studies only one study per intervention was identified; brief strategic therapy vs. CBT (Castelnuovo et al., 2011), adapted motivational interviewing combined with self-help vs. self-help (Cassin et al., 2008), self-help combined with treatment as usual (TAU) vs. TAU (Grilo et al., 2013), shape exposure CBT vs. cognitive restructuring CBT (Hilbert & Tuschen-Caffier, 2004), CBT combined with low calorie diet (Masheb, Grilo & Rolls, 2011), CBT vs. CBT in group format (Ricca et al., 2010) and finally a study with three conditions; CBT vs. schema therapy vs. Appetite focused CBT (McIntosh et al., 2016).

In Carter & Fairburn (1998), only the CBT-gsh vs. wait list was included (the third condition, self-help, was not included in any meta-analysis). The CBT vs. BWL comparison from the Grilo et al. (2011) study was included while the third arm (CBT + BWL) was not included in any meta-analysis. In Kelly & Carter (2014) the self-help meal planning combined with self-compassion therapy arm was not included. However, we included the CBT-gsh vs. wait list comparisons.

Overall evidence quality

The GRADE method was used and the majority of studies were downgraded two levels. The two most common reasons for downgrading were precision and study quality. Only four meta-analyses received moderate rating, which was the highest level of quality in this review assigned to any single study. The majority of studies received limited evidence of quality. Please see Table 1 for details.

Table 1 Effect size and quality grade of included studies along with the reason for reduction of quality grade.

Outcome/intervention	k Studies, n participants k, n	Effect size (95% CI)	Overall effect Z (p)	Intervention length: weeks	Quality GRADE	Reason for reduction in quality grade	
SSRI vs. Placebo	6, 285			6–16			
Remission	6, 264	RD = 0.15 (0.02: 0.27)	2.34 (0.02)SSRI*		⊕⊕ΟΟ	−1 Bias
−1 Imprecision	
BE frequency	6, 257	SMD = −0.45 (−0.82: −0.09)	2.43 (0.02)SSRI		⊕⊕ΟΟ	−1 Bias
−1 Imprecision	
ED psychopathology	1, 54	n.a.		⊕ΟΟΟ	One small study***	
BMI	5, 237	SMD = 0.01 (−0.39: 0.41)	0.04 (0.97)		⊕ΟΟΟ	−1 Bias, inconsistency
−2 Precision	
Depressive symptoms	4, 148	SMD = −0.16 (−0.49: 0.16)	0.98 (0.33)		⊕ΟΟΟ	−2 Imprecision
−1 Bias	
Lisdexamphetamine (LDX) vs. placebo	3a, 850			11 and 12			
Remission	3, 850	RD = 0.25 (0.19: 0.31)	8.60 (<0.001)LDX		⊕⊕ΟΟ	−1 Bias
−1 Indirectness	
BE frequency	3, 849	SMD = −0.76 (−0.99: −0.53)	6.43 (<0.001)LDX		⊕⊕ΟΟ	−2 Bias, indirectness, imprecision	
BMI	3, 852	SMD = −5.23 (−6.52: −3.94)	7.93 (<0.001)LDX		⊕⊕ΟΟ	−1 Bias
−1 Indirectness	
Depressive symptoms	1, 120	Data could not be extracted			⊕ΟΟΟ	One small study***	
Anti-convulsive vs. placebo	2, 445			16			
Remission	2, 443	RD = 0.05 (−0.47: 0.56)	n.a		⊕ΟΟΟ	−1 Inconsistency
−2 Imprecision	
BE frequency	2, 445	SMD = −0.28 (−2.34: 1.77)	n.a		⊕ΟΟΟ	−1 Inconsistency
−2 Imprecision	
ED psychopathology	1, 51	n.a.		⊕ΟΟΟ	One small study***	
BMI	2, 445	Combining outcome measures was not possible			⊕ΟΟΟ	−2 Imprecision
−1 Indirectness	
Depressive symptoms	2, 445	Combining outcome measures was not possible			⊕ΟΟΟ	−2 Imprecision
−1 Indirectness	
Anorexiants vs. placebo	2, 103	Combining outcome measures was not possible		Six months	⊕ΟΟΟ	One small study***	
Drugs & psychol. treatment combined	5, –	Only one study per combination of drug and psychological treatment	⊕ΟΟΟ	One small study***	
CBT vs. waitlist				8–20			
Remission	4, 272	RD = 0.40 (0.30: 0.50)	7.83 (<0.001)CBT		⊕⊕⊕Ο	−1 Imprecision	
BE frequency	4, 272	SMD = −0.83 (−1.11: −0.55)	5.76 (<0.001)CBT		⊕⊕ΟΟ	−1 Imprecision
−1 Indirectness	
ED psychopathology	4, 269	MD = −0.50 (−0.88; −0.12)	2.56 (0.01)CBT		⊕⊕ΟΟ	−1 Bias
−1 Imprecision	
BMI	3, 220	SMD = −0.09 (−0.55: 0.37)	0.40 (0.69)		⊕ΟΟΟ	−1 Inconsistency
−2 Imprecision	
Depressive symptoms	4, 267	SMD = −0.42 (−0.67: −0.18)	3.40 (<0.001)CBT		⊕⊕⊕Ο	−1 Imprecision	
CBT-gsh vs. waitlist	8, ca. 400			3–24			
Remission	6, 333	RD = 0.25 (0.12: 0.38)	3.67 (<0.001)CBT-gsh		⊕⊕⊕Ο	−1 Imprecision, bias	
BE frequency	7, 358	SMD = −0.51 (−0.84: −0.17)	2.98 (0.003)CBT-gsh		⊕⊕ΟΟ	−1 Imprecision
−1 Inconsistency, bias	
ED psychopathology	6, 348	SMD = −0.58 (−0.98: −0.17)	2.81 (0.005)CBT-gsh		⊕⊕⊕Ο	−1 Imprecision	
BMI	7, 384	SMD = −0.52 (−2.44: 1.40)	0.53 (0.60)		⊕ΟΟΟ	−1 Imprecision
−2 Inconsistency	
Depressive symptoms	5, 282	SMD = −0.35 (−0.63: −0.07)	2.48 (0.01)CBT-gsh		⊕⊕ΟΟ	−1 imprecision
−1 Bias, inconsistency	
IPT vs. CBT/CBT-gsh	2, 303			20–24			
Remission	2, 265	RD = −0.02 (−0.13: 0.09)	0.43 (0.67)		⊕⊕ΟΟ	−1 Imprecision
−1 Indirectness	
Remission 12 months	2, 265	RD = −0.00 (−0.12: 0.12)	0.07 (0.95)		⊕⊕ΟΟ	−1 Imprecision
−1 Indirectness	
BE frequency	2, 299	SMD = 0.08 (−0.15: 0.31)	0.69 (0.49)		⊕⊕ΟΟ	−2 Imprecision	
BE frequency 12 months	2, 279	SMD = −0.04 (−0.27: 0.20)	0.31 (0.76)		⊕⊕ΟΟ	−2 Imprecision	
ED psychopathology	1, 141	n.a.		⊕ΟΟΟ	One small study***	
BMI	2, 299	MD = −0.25 (−1.39: 0.90)	0.43 (0.67)		⊕⊕ΟΟ	−2 Imprecision	
BMI 12 months	2, 279	MD = −0.33 (−1.55: 0.89)	0.54 (0.59)		⊕⊕ΟΟ	−2 Imprecision	
Depressive symptoms	1, 158	n.a.		⊕ΟΟΟ	One small study***	
BWL vs. CBT**	4, 375			12–20			
Remission	4, 375	RD = −0.06 (−0.22: 0.11)	0.64 (0.52)		⊕ΟΟΟ	−1 Inconsistency
−2 Imprecision	
Remission 12 months	3, 300	RD = −0.13 (−0.25: −0.02)	2.35 (0.02)CBT		⊕⊕ΟΟ	−2 Imprecision	
BE frequency	4, 375	SMD = 0.27 (0.05: 0.48)	2.40 (0.02)CBT		⊕⊕ΟΟ	−2 Imprecision	
BE frequency 12 months	3, 300	SMD = 0.24 (0.01: 0.46)	2.03 (0.04)CBT		⊕⊕ΟΟ	−2 Imprecision	
ED psychopathology	1, 139	n.a.		⊕ΟΟΟ		
BMI	4, 376	SMD = −1.07 (−2.40: 0.25)	1.59 (0.11)		⊕ΟΟΟ	−2 Inconsistency
−1 Imprecision	
BMI 12 months	3, 300	SMD = −0.23 (−1.46: 0.99)	0.37 (0.71)		⊕⊕ΟΟ	−1 Inconsistency
−1 Imprecision	
Depressive symptoms	3, 222	MD = 1.03 (−1.20: 3.25)	0.91 (0.37)		⊕⊕ΟΟ	−2 Imprecision	
Depressive symptoms 12 months	2, 133	MD = 0.25 (−2.53: 3.03)	0.18 (0.86)		⊕ΟΟΟ	−2 Imprecision
−1 Indirectness	
Notes:

The number of studies for each comparison, along with number of participants, effect sizes, intervention length, and quality of the studies are presented in detail.

CBT, cognitive behavior therapy; BE, binge eating; BMI, body mass index; BWL, behavior weight loss; ED, eating disorder; -gsh, guided self-help; IPT, interpersonal psychotherapy; MD, mean difference; SMD, standardized mean difference; SSRI, selective serotonin reuptake inhibitor; RD, risk difference.

Degree of evidence; ⊕⊕⊕⊕ = High, ⊕⊕⊕Ο = Moderate, ⊕⊕ΟΟ = Low, and ⊕ΟΟΟ = Insufficient (Very low).

a One publication involved a multicenter study.

* Superscript indicate the favored treatment.

** See manuscript for the different formats involved.

*** The confidence in the estimate/evidence is very low (⊕ΟΟΟ) when the evidence is based on only one small study (i.e., −2 points for imprecision and −1 point for indirectness).

Meta-analyses

In the following, we present the summary of the meta-analyses (please see Table 1 for detailed information on each meta-analysis). The results from the RD analyses should be interpreted as per this example; RD = 0.15 [0.02; 0.27] means that for every set of 1,000 participants receiving treatment, there was 150 (95% CI [20–270]) more remitted persons among those who received the treatment compared to those in the control condition. The SMD can be basically interpreted as Cohen’s d (Cohen, 1988) with values of 0.2, 0.5, and 0.8 marking small, moderate and large effects, respectively. Regarding side effects, only significantly reported differences between groups are reported in this summary.

SSRI vs. placebo

SSRI vs. placebo was investigated in six studies (Grilo et al., 2012; Grilo, Masheb & Wilson, 2005; Guerdjikova et al., 2008; Hudson et al., 1998; McElroy et al., 2000, 2003; Pearlstein et al., 2003). Participants’ age were 18–60 years, mostly women (>70%) with weight >85% of ideal body weight. Treatments ranged between 6–16 weeks and only one study reported six and 12 months follow ups (Grilo, Masheb & Wilson, 2005; Grilo et al., 2012). Flexible dosage and titration was used in all studies but one (Grilo, Masheb & Wilson, 2005).

Remission was reported in all six studies (n = 264). The RD = 0.15 was in favor of SSRI treatment compared with placebo at the end of the treatment. Binge eating frequency was reported in all six studies (n = 257), and comparisons resulted in SMD = −0.45 in favor of SSRI treatment compared with placebo at the end of the treatment. Eating disorder psychopathology was only reported in one study comparing SSRI vs. placebo (Grilo, Masheb & Wilson, 2005). BMI was reported in five studies (n = 237) (Grilo, Masheb & Wilson, 2005; Guerdjikova et al., 2008; Hudson et al., 1998; McElroy et al., 2000, 2003). There was no significant difference for SSRI vs. placebo. Symptoms of depression were reported in four studies (n = 148) (Grilo, Masheb & Wilson, 2005; Guerdjikova et al., 2008; McElroy et al., 2003; Pearlstein et al., 2003) showing no significant difference between SSRI and placebo.

Side effects

Hudson et al. (1998) reported significantly more insomnia, nausea, and abnormal dreams in the fluoxetine condition compared with placebo. McElroy et al. (2003) reported significantly more fatigue and perspiration in the citalopram group compared with the placebo group. There were significantly more participants in the sertraline group who reported insomnia (McElroy et al., 2000). One study reported 1.3–4 times more sedation, nausea, dry mouth, and decreased libido (Pearlstein et al., 2003).

Lisdexamfetamine vs. placebo

One research group had two publications (n = 850) (McElroy et al., 2015c, 2015d) including a multicenter study with participants from Holland, Spain, and Sweden (McElroy et al., 2015c). Hence, the results from three samples are combined for this meta-analysis. Participants were 18–55 years, mostly women >81% with a BMI range of 18–45. Treatment length was 11 or 12 weeks. One study (McElroy et al., 2015d) investigated the effect of flexible dosages (30–70 mg/day), while the other one (Guerdjikova et al., 2016) administered a standardized dosage (70 mg/day). The outcomes are based on those who received 70 mg/day. The studies did not report any follow up data.

Remission was reported in three studies (n = 850). The RD = 0.25 was in favor of lisdexamfetamine compared to those receiving placebo at the end of treatment. Binge eating frequency was reported in all three studies (n = 849). There was a significant difference in favor for lisdexamfetamine compared with placebo (Table 1). Eating disorder psychopathology was not reported. BMI: Three studies reported weight difference from baseline (n = 852), and the result (SMD = −5.23) favored the lisdexamfetamine condition compared with placebo. Symptoms of depression were reported from one study only (McElroy et al., 2015c).

Side effects

In both publications, a majority of participants in the lisdexamfetamine group (>80%) and the placebo group (>50%) reported side effects. In the placebo group, 0–3% discontinued due to side effects compared with 3–6% in the lisdexamfetamine group. One participant died due to methamphetamine and amphetamine toxicity (McElroy et al., 2015d).

Anti-convulsive vs. placebo

Two studies investigated the effect of anti-convulsive medication for their mood stabilization characteristics vs. placebo (n = 443) (Guerdjikova et al., 2009; McElroy et al., 2007). Participants were between 18 and 65 years with BMI > 30 (mean BMI > 38.5) and the majority were women (>80%). The treatment in both studies was 16 weeks with no follow-ups. Both studies used flexible dosage of lamotrigin (Guerdjikova et al., 2009) and topiramate (Guerdjikova et al., 2009; McElroy et al., 2007).

Remission: The outcome in terms of remission was inconsistent between the two studies (n = 443), resulting in a non-significant outcome. Binge eating frequency was inconsistent (n = 445) between the two studies, resulting in a non-significant outcome (Table 1). Eating disorder psychopathology was only reported in one study (Guerdjikova et al., 2009). BMI: only one study reported data in a proper format for inclusion in a meta-analysis (McElroy et al., 2007). Symptoms of depression were investigated in both studies but only one study reported the mean and thus, no meta-analysis was performed.

Side effects

In the topiramate study (McElroy et al., 2007), significantly more participants in the intervention condition compared to placebo reported the following adverse events: paresthesia, upper respiratory tract infection, taste perversion, difficulty with concentration/attention, and difficulty with memory not otherwise specified (McElroy et al., 2007).

Anorexiants vs. placebo

Two studies (n = 103) were identified. They investigated chromium picolinate with two dosages (800–1,000 μg/day) (Brownley et al., 2013) and Orlistat (120 mg/day) (Golay et al., 2005). Participants were 18–65 years (mean ages were 37 and 41) and they were mostly women (>83%) with a mean BMI of 37 and 43, respectively. One study included overweight and obese (Brownley et al., 2013) while the other included only obese participants. The duration of the treatments were six months in both studies with no follow-ups.

The outcome measures were presented in different formats between the two studies such that no meta-analysis was performed. Remission: Only one study reported results on remission (Golay et al., 2005). There was no significant difference between the treatment and the placebo group. Binge eating frequency was reported in both studies but with no significant differences between treatment and placebo in both studies. Eating disorder psychopathology was reported in one study (Brownley et al., 2013) with no significant differences between treatment groups and placebo. BMI: weight reduction was significantly larger for the orlistat group compared with placebo (Golay et al., 2005). Symptoms of depression: there were no significant effects in either study. Quality of life was presented in one study with no significant effect (Golay et al., 2005).

Side effects

No significant differences between groups were reported in the chromium picolinate study (Brownley et al., 2013) and no side effects were reported in the orlistat study (Golay et al., 2005).

Drug and psychological treatment combined

The five studies could not be combined for a meta-analysis (Agras et al., 1994; Claudino et al., 2007; Grilo, Masheb & Salant, 2005, Grilo, Masheb & Wilson, 2005; Grilo & White, 2013). The following combinations were investigated: A combination of desipramine, weight loss, and CBT vs. CBT combined with weight loss program (Agras et al., 1994); topiramate combined with CBT vs. CBT (Claudino et al., 2007); combination of orlistat, dietary advice, and CBT vs. placebo, dietary advice, and CBT (Grilo, Masheb & Salant, 2005); a combination of fluoxetine and CBT vs. placebo and CBT (Grilo, Masheb & Wilson, 2005); and a combination of orlistat and BWL vs. placebo and BWL (Grilo & White, 2013). Participants were 18–65 years old, >78% were women, with BMI ≥ 27. Treatments were 12–36 weeks and followed up at three months (Agras et al., 1994; Grilo, Masheb & Salant, 2005), six months (Grilo & White, 2013), and at 12 months (Grilo et al., 2012).

Side effects

Two studies did not report on side effects (Agras et al., 1994; Grilo, Masheb & Wilson, 2005). Those who received topiramate reported significantly more paresthesia, taste perversion, dysuria, and leg pain compared with the placebo group. However, the placebo group reported significantly more insomnia (Claudino et al., 2007). Those who received orlistat reported more gastrointestinal events that resolved (Grilo, Masheb & Salant, 2005; Grilo & White, 2013). However, two participants dropped out due to these side effects (Grilo, Masheb & Salant, 2005).

CBT vs. wait list

CBT vs. wait list was investigated in four studies (Dingemans, Spinhoven & van Furth, 2007; Grilo, Masheb & Wilson, 2005; Peterson et al., 2009; Schlup et al., 2009). Participants were mostly women (>70%), 18–65 years old, with a mean age of 43.5 years and a mean BMI of 36.6. Treatments ranged between eight (Schlup et al., 2009), 16 (Grilo, Masheb & Wilson, 2005), and 20 weeks (Dingemans, Spinhoven & van Furth, 2007; Peterson et al., 2009), and the number of sessions ranged between eight and 16. No follow up data were presented. The effects were investigated in individual format in one study (Grilo, Masheb & Wilson, 2005), while the other three studies used group format (Dingemans, Spinhoven & van Furth, 2007; Peterson et al., 2009; Schlup et al., 2009). The content of the interventions was assessed as fairly equal despite different formats, number of sessions, and the duration of treatment.

Remission: Four studies reported remission (n = 272) in favor of CBT compared with wait list, RD = 0.40. Binge eating frequency was reported in all four studies (n = 272), however due to different frequency assessments (BE episodes (Grilo, Masheb & Wilson, 2005; Schlup et al., 2009), and BE days during the last 28 days (Dingemans, Spinhoven & van Furth, 2007; Peterson et al., 2009)) the results in the meta-analysis are presented as SMD. The results were in favor of CBT compared with wait list, SMD = −0.83. Eating disorder psychopathology was investigated with the EDE-Q and presented in four studies (n = 269) (Dingemans, Spinhoven & van Furth, 2007; Grilo, Masheb & Wilson, 2005; Peterson et al., 2009; Schlup et al., 2009). The results favor the CBT intervention compared with the wait list, SMD = −0.50. BMI: Three of the four studies reported data for inclusion in the meta-analysis (n = 220) on BMI change (Grilo, Masheb & Wilson, 2005; Peterson et al., 2009; Schlup et al., 2009). No significant effect was found (Table 1). The fourth study did not show any significant results either (Dingemans, Spinhoven & van Furth, 2007). Symptoms of depression were reported in four studies (n = 267) (Dingemans, Spinhoven & van Furth, 2007; Grilo, Masheb & Wilson, 2005; Peterson et al., 2009; Schlup et al., 2009). Different outcome scales were used in the studies but were synthesized for this meta-analysis showing a result in favor of the CBT compared with the wait list condition, SMD = −0.42.

Side effects were not reported in any of the studies.

CBT self-help vs. wait list

There were eight studies investigating CBT-gsh vs. wait list (Carrard et al., 2011; Carter & Fairburn, 1998; Grilo & Masheb, 2005; Grilo et al., 2014; Kelly & Carter, 2014; Masson et al., 2013; Shapiro et al., 2007; Ter Huurne et al., 2015). Participants were 18 years and older and 87% were women. Two studies included only overweight or obese participants (BMI ≥ 27) (Grilo & Masheb, 2005; Shapiro et al., 2007) and one study included only obese participants (Grilo et al., 2014). The remaining studies had no BMI inclusion criteria. The number of treatment weeks varied between three (Kelly & Carter, 2014), 10–13 (Carter & Fairburn, 1998; Grilo & Masheb, 2005; Masson et al., 2013; Shapiro et al., 2007), 15–18 (Ter Huurne et al., 2015), 16 (Grilo et al., 2014), and 24 (Carrard et al., 2011). There were five studies (Carrard et al., 2011; Carter & Fairburn, 1998; Grilo & Masheb, 2005; Grilo et al., 2014; Ter Huurne et al., 2015) that investigated eating disorder specific CBT, one study investigated DBT (Masson et al., 2013) and one study investigated a treatment with focus on healthy eating and weight control with CBT) (Shapiro et al., 2007). The degree of support varied between the studies: it ranged between no support (Carter & Fairburn, 1998; Grilo et al., 2014), only one session before the program started (Kelly & Carter, 2014), one phone call (Shapiro et al., 2007), 20 min telephone support biweekly (Masson et al., 2013), mail correspondence twice a week (Ter Huurne et al., 2015), and weekly mail contact + automated feedback from the program (Carrard et al., 2011) and six brief biweekly meetings (Grilo & Masheb, 2005). First we present the results on all eight studies, followed by the results on the four studies that provided adequate and reasonable dose of support (Carrard et al., 2011; Grilo & Masheb, 2005; Masson et al., 2013; Ter Huurne et al., 2015).

Remission was presented in six studies (n = 333) (Carrard et al., 2011; Carter & Fairburn, 1998; Grilo & Masheb, 2005; Grilo et al., 2014; Masson et al., 2013; Shapiro et al., 2007) and the result was in favor of self-help compared with wait list, RD = 0.25. Binge eating frequency was reported in seven studies (n = 358) (Carrard et al., 2011; Carter & Fairburn, 1998; Grilo & Masheb, 2005; Grilo et al., 2014; Kelly & Carter, 2014; Masson et al., 2013; Shapiro et al., 2007) and the result was in favor of self-help vs. wait list at end of treatment, SMD = −0.51. Eating disorder psychopathology was reported with the EDE-Q in six studies (n = 348) (Carrard et al., 2011; Carter & Fairburn, 1998; Grilo et al., 2014; Kelly & Carter, 2014; Masson et al., 2013) and the result was in favor of self-help, MD = −0.58. BMI was reported in seven studies (n = 384) (Carrard et al., 2011; Carter & Fairburn, 1998; Grilo & Masheb, 2005; Grilo et al., 2014; Kelly & Carter, 2014; Shapiro et al., 2007; Ter Huurne et al., 2015) and the result showed no significant difference between self-help vs. wait list. Symptoms of depression was reported in five studies (n = 282) (Carrard et al., 2011; Grilo & Masheb, 2005; Grilo et al., 2014; Kelly & Carter, 2014; Ter Huurne et al., 2015) and the result was in favor of the self-help compared with wait list (Table 1), SMD = −0.31, albeit a small and clinically not a meaningful effect.

The results for the four studies with reasonable dose of support Remission and binge eating frequency was presented in three studies (n = 192) (Carrard et al., 2011; Grilo & Masheb, 2005; Masson et al., 2013) and the result was in favor of self-help compared with wait list, RD = 0.32 (95% CI [0.20; 0.43]) and SMD = −0.56 (95% CI [−0.86; −0.26]), respectively. Eating disorder psychopathology was reported with the EDE-Q in three studies (n = 219) (Carrard et al., 2011; Masson et al., 2013; Ter Huurne et al., 2015) and the result was in favor of self-help, MD = −0.56 (95% CI [−0.85; −0.28]). BMI was reported in three studies (n = 220) (Carrard et al., 2011; Grilo & Masheb, 2005; Ter Huurne et al., 2015) and the result showed no significant difference between self-help vs. wait list, MD = −0.73 (95% CI [−3.93; 2.48]). Symptoms of depression was reported in three studies (n = 211) (Carrard et al., 2011; Grilo & Masheb, 2005; Ter Huurne et al., 2015) and the result was in favor of the CBT-gsh compared with wait list, SMD = −0.35 (95% CI [−0.63; −0.07]), albeit a small and clinically not a meaningful effect.

Two studies reported follow-ups, one at two months (Shapiro et al., 2007), and at one year (Grilo et al., 2014). There is insufficient evidence for long-term evaluation.

Side effects were not reported in any of the studies.

IPT vs. CBT

Two studies investigated IPT vs. CBT or CBT-gsh (Wilfley et al., 2002; Wilson et al., 2010) with long term follow ups (Hilbert et al., 2012, 2015). Participants were 18 years and older, mean ages were 45 and 49 years, the majority were women (79% and 83%), and with a BMI range of 27–48 (Wilfley et al., 2002) and up to 45 (Wilson et al., 2010), with a mean BMI of 36 and 37, respectively. Treatments were 20 weekly group sessions plus three individual sessions for IPT and CBT (Wilfley et al., 2002) and the other study had 19 individual sessions of IPT vs. CBT-gsh with ten 25-min sessions over 24 weeks (Wilson et al., 2010). Both studies reported 12 month follow up data that was included in meta-analyses while longer term follow up showed too high rate of drop out in one study (>50%) (Wilfley et al., 2002).

Remission was reported in both studies at end of treatment (n = 265), and at 12 months follow-up (n = 265). The treatments were equivalent at end of treatment, and at 12 months follow up (Table 1). Binge eating frequency was reported in both studies at end of treatment (n = 299) and at 12 months follow up (n = 279). Both treatments were equal at both occasions. Eating disorder psychopathology was only reported in one study (Wilson et al., 2010), hence no meta-analyses was performed. BMI was reported in both studies at end of treatment (n = 299) and at 12 months follow up (n = 279). IPT and CBT were equal at both occasions. Symptoms of depression were only reported in one study (Wilfley et al., 2002) and thus there was insufficient evidence for a meta-analysis.

Side effects were not reported in any of the studies.

BWL vs. CBT

Four studies (n = 375) investigated the effect of BWL vs. CBT (Grilo & Masheb, 2005; Grilo et al., 2011; Munsch et al., 2007; Wilson et al., 2010). Participants were 18 years and older, the majority were women (63–89%) with a BMI ≥ 27. One study included participants with a BMI between 30 and 55 (Grilo et al., 2011). Two studies compared the effect of BWL and CBT, both in group format, over 16 sessions during 16 weeks (Munsch et al., 2007) and 24 weeks (Grilo et al., 2011). One study investigated 24 weeks of BWL (20 sessions) with CBT-gsh including 10 sessions of 25 min long support (Wilson et al., 2010). One study compared CBT-gsh vs. BWL-gsh with biweekly 20-min sessions during 12 weeks (Grilo & Masheb, 2005). Three studies investigated long-term follow-ups (Grilo et al., 2011; Munsch et al., 2007; Wilson et al., 2010).

Remission was reported in all four studies at end of treatment (n = 375) and by three studies at one-year follow up (n = 300) (Grilo et al., 2011; Munsch et al., 2007; Wilson et al., 2010). At end of treatment there was no significant difference between BWL and CBT (group/CBT-gsh), but at one year follow up there was a significant difference in favor of CBT (group/CBT-gsh) compared with BWL, RD = −0.13. Binge eating frequency was reported in all four studies at end of treatment (n = 375) and reported in three studies at one year follow up (n = 300) (Grilo et al., 2011; Munsch et al., 2007; Wilson et al., 2010). At end of treatment and at one year follow up there were significant difference in favor of the CBT conditions, SMD = 0.27, and SMD = 0.24, respectively. Eating disorder psychopathology was only reported in one study (Wilson et al., 2010). BMI was reported in all four studies at end of treatment (n = 376) and in three studies at the one-year follow up (n = 300) (Grilo et al., 2011; Munsch et al., 2007; Wilson et al., 2010). We found no significant difference between BWL and CBT at any of the time points. Symptoms of depression were reported in three studies at end of treatment (n = 222) (Grilo & Masheb, 2005; Grilo et al., 2011; Munsch et al., 2007), and in two studies at one-year follow up (n = 133) (Grilo et al., 2011; Munsch et al., 2007). We did not find any significant differences between BWL and CBT at end of treatment or at the one-year follow up.

Side effects were not reported in any of the studies.

Discussion

The aim of this systematic review was to evaluate the efficacy and the quality of evidence of psychological, pharmaceutical and combined treatments for BED, and potential side effects. We found moderate-quality evidence for remission (i.e., discontinuation of binge eating) for the comparisons CBT vs. wait list and CBT-gsh/self help (sh) vs. wait list. CBT was associated with 400 more per 1,000 in remission and CBT-gsh/sh 250 more per 1,000 compared with wait list. Moderate-quality evidence was also found for reduction of eating disorder specific psychopathology for CBT-gsh/sh vs. wait list. The IPT and CBT comparisons had low-quality evidence for remission, BE frequency, and weight loss. However, they were equally efficacious at end of treatment and at one-year follow-up for these outcomes.

With regard to pharmaceutical treatments, SSRIs and lisdexamfetamine resulted in remission and decreased frequency of binge eating episodes at the end of treatment with low quality evidence. However, the long-term effect of pharmacotherapy is largely unknown. Interestingly, the effect of SSRIs on depressed mood in BED was not significant despite SSRIs being anti-depressant medications. Lisdexamfetamine was the only intervention showing some positive effect on BMI, a finding that is expected and encouraging, as partial loss of appetite is a known side effect of this category of drugs.

The remission rate at post-treatment was thus highest for CBT, followed by CBT-gsh, Central Nervous System stimulants, and SSRI. The effects of IPT were similar to those of CBT, although the quality of the evidence in that regard was low. However, it should be noted that psychological treatments in most studies have been compared to a wait list control group while pharmaceutical treatments are generally compared to a placebo condition. Given the difference in the nature of the control group in psychological vs. pharmaceutical studies, and lack of head-to-head studies, any indirect comparisons should be interpreted with caution.

Available recent systematic reviews of treatment of DSM-IV/DSM-5 BED published in the last decade were scrutinized with focus on RCTs (please see the Introduction). Overall, the general findings were similar across reviews, but there is currently no data to allow evaluation of longer-term effects of pharmacotherapy-only treatment for BED. In terms of limitations, the majority of reviews mention the inconsistencies of outcomes measures across trials, the lack of long-term follow-ups, especially in trials on drug treatment of BED, and methodological shortcomings that lead to exclusion of a significant number of trials from systematic reviews and meta-analyses.

The present review and meta-analyses confirms most of the conclusions from previous reviews. The majority of studies did not report long-term follow-up. Lack of information in pharmaceutical studies about the outcome and use or discontinued use of drugs after post-treatment was striking. For CBT, CBT-gsh, IPT, and BWL, follow-ups were available up to one year after the end of the treatment in a few studies with low risk of bias. However, even fewer studies reported longer outcome than one year post treatment, and usually with low quality and focusing on very few variables. Importantly, we observed lack of data on the specific psychopathology of BED and quality of life, even though patients with BED are known to have reduced quality of life. Only two studies included a measure of quality of life (Golay et al., 2005; Ricca et al., 2010). This is an important area for future research, together with health economic studies, which are also lacking.

Despite a fair total number of studies investigating the efficacy of treatments for BED in general, not many studies met the fairly generous inclusion criteria and survived the check upon the few exclusion criteria to be included in the current systematic review. In addition, this review clearly shows that a mentionable number of studies have relatively small samples, thus imposing limitation to reach at robust pooled data to investigate some of the specific research questions of interest. As an example, we found too few studies investigating the efficacy of several drugs such as NDRI and SNRI, and psychological treatments such as schema therapy to run meta-analyses. In some cases, such as for studies on anorexiants, we found only two studies that could not be merged due to lack of adequate data. Attempts to retrieve necessary data for our analyses from the authors was met with no response.

The overall quality of the included studies was high, and limitations such as small sample size were weighted by withdrawing points from the overall rating of these studies. Lack of clarity in some aspects of the procedure (e.g., details of randomization) in several studies resulted in adjustment of the quality rating of such studies. As an example, in some studies the allocation of participants was not described adequately (i.e., to provide transparency) or, in some studies, not at all. In other studies, it was unclear to what extent the assessors were blinded to the allocation of the participants at post-treatment or at follow-up, which resulted in adjustment of the quality rating. Another indication of good quality of the included studies is use of structured or semi-structured interviews for establishing the diagnosis of BED. The exclusion criteria in the studies that were included into the current meta-analyses were logical and reflective of everyday clinical praxis (e.g., excluding those with psychosis or acute suicidality to make sure they receive adequate treatment), and thus not contributing to create limitation, or bias in terms of generalizability of the outcome of the current meta-analysis.

The majority of the participants were adult females with BED and with concurrent overweight or obesity, recruited via ads, or in some studies through referrals from different clinics. It is noteworthy that we found no studies that included adolescents. The extent to which the conclusions of the current review can be generalized to other populations (men, adolescents, or minority groups) is limited. However, the severity of the symptoms, and the comorbid psychopathology of the included participants indicate that the included total sample in the current review is not markedly different from those seeking treatment from specialist psychiatry and other relevant caregivers. In all the included studies, participants suffer from full or sub-threshold BED based on the DSM-IV (American Psychiatric Association, 1994), DSM-IV-TR (American Psychiatric Association, 2000), or other equivalent diagnostic systems. The majority of those with a sub-threshold BED diagnosis according the above mentioned diagnostic manuals would meet the full BED diagnosis according to the most recent version of the DSM (i.e., DSM-5: American Psychiatric Association, 2013) given the changes made in DSM-5 with regard to duration and frequency of binge eating episodes.

The length of treatments varied substantially within both pharmaceutical and psychological treatments, but in the majority of the included studies, a clinically adequate dose of intervention was delivered. Titration and optimization of pharmaceutical treatments seem to become more common in newer studies, making them ecologically more valid. The length of most of the psychological treatments varied between 10 and 24 weeks, but a few studies had considerably shorter duration (e.g., a self-help treatment that was only three weeks long). The treatment content among the included psychological treatment studies varied substantially as well. As several essential components of CBT for ED were missing in some treatment packages in several trials, they could not be combined with standard CBT treatment in other studies in a meaningful way, and were thus not entered into any meta-analysis. This led to inclusion of only four comparisons between CBT and wait list.

Compliance is an important factor in the interpretation of outcome of treatment trials. For pharmaceutical trials, compliance was fairly high, but in some of the important multicenter studies, the compliance was not higher than 50% due to side effects of medication, early drop-out, divergence from the trial protocol, lack of effects, or other reasons. Improved procedures in pharmaceutical studies such as titration and regular checks increase the safety of the participants, transparency, and fidelity. Lack of compliance was easier to detect in newer well-controlled studies, compared to earlier studies. For psychological treatments, compliance was reported in some of the studies by defining good compliance in terms of attending a specific number of sessions in face-to-face treatments, or completing a specific number of modules in self-help treatments, while other studies did not report a clear picture of compliance. In studies where compliance was reported, it ranged between 60% and 93%, and although no specific and statistically robust and reliable pattern emerged, compliance seemed to be higher in face-to-face treatments compared to guided self-help treatments.

The drop-out rate in the studies included in this meta-analysis was acceptable as a direct consequence of the inclusion and exclusion criteria of the current study. Treatment studies with high rate of drop-out (i.e., >30%) were deemed too biased to be included in the meta-analysis. We also noted a tendency toward using more advanced and adequate statistical analysis (e.g., multilevel modeling) in more recent studies that more efficiently handle the drop-out and provide intention to treat analysis per default. Half of the studies investigating psychological treatments reported and investigated the drop-out in light of demographic factors and outcome variables. Lower socio-economic status and higher severity of symptoms seemed to be related to higher drop-out, but it was not systematically investigated in the current study.

Two other interesting observations were made during the analyses. The first one was lack of an a priori power analysis in a marked number of included studies, especially the older studies. The second one was vague and general statements regarding conflict of interest in some of the pharmaceutical studies with regard to the relationship between those that conducted the study and the pharmaceutical companies supplying the drugs.

With respect to pharmacological treatments, the adverse effects associated with SSRI and lisdexamfetamine when described for other disorders (Frampton, 2016; Kostev et al., 2014) were also present for participants with BED. The risk for adverse events from psychological treatments is largely unknown; the included studies did not report any adverse events but it is unclear whether this was systematically investigated. In future studies, systematic screening and report of adverse effects should be part of psychological treatment trials as well.

In terms of the clinical implications, our findings would add support to the current guidelines such as NICE (National Institute for Health and Care Excellence, 2017) and the Royal Australian and New Zealand College of Psychiatrists clinical practice guidelines for the treatment of eating disorders (Hay et al., 2014). A recent review of nine evidence-based clinical guidelines (Hilbert, Hoek & Schmidt, 2017) showed that current guidelines endorse the main empirically supported treatments with considerable agreement, but they differ markedly in additional recommendations. The NICE guidelines do not suggest use of pharmacotherapy as the sole treatment of BED. Lack of research on the long-term efficacy of for example SSRI or lisdexamfetamine in our review adds support to the recommendation of NICE when it comes to pharmacotherapy. CBT-based treatments (in guided self-help-, group-, or individual format) are suggested as the first line treatment of choice. IPT might be employed if CBT is not producing effects, or if the patient does not wish to receive CBT. This recommendation is also in line with our findings, indicating limited scientific evidence of IPT.

Several limitations of the current meta-analysis and review should be mentioned to provide a framework for adequate interpretations. In the planning phase of the present study, we discussed running separate analysis for treatment of BED with, and without obesity. It turned out that the majority of participants in the studies had significant overweight or obesity. The number of studies meeting the criteria for the present meta-analysis was too limited to allow separate analysis with regard to weight status. Access to raw data from several studies and data aggregation might provide a very valuable source for investigating the outcome of different treatments for BED in relation to weight status of the participants. Choice of inclusion and exclusion variables and how they are defined on a scale from conservative to liberal does affect the outcome of any review and meta-analysis. Although our criteria might be considered to belong to the more conservative side, we believe that clear-cut and fairly conservative criteria are necessary to identify studies with sound methodology to make valid inferences. This is, however, a pre-analytic assumption, and the choice and definition of criteria for inclusion and exclusion can always be discussed. This may per se justify the need for replication of not only original studies, but also reviews and meta-analyses. Including studies with high risk of bias may introduce too much uncertainty to make significant gains from including a larger number of studies in a review. As an example, studies with high level of attrition might produce highly biased outcomes that are systematically related to the pattern of attrition. Empirical investigation of a liberal vs. a conservative approach in systematic reviews might be highly informative for future decisions about the choice of inclusion and exclusion criteria.

In conclusion, we found moderate support for the efficacy of CBT and CBT-gsh (with moderate quality of evidence), and modest support for IPT, SSRI and lisdexamfetamine (with low quality of evidence) in the treatment of adults with BED in terms of cessation of or reduction in the frequency of binge eating. It should be noted that males and adolescents were underrepresented in the included studies. Lisdexamfetamine was the only treatment that showed a clinically non-significant and very modest effect on weight loss (with low quality of evidence). While there is limited support for the long-term effect of psychological treatments, we have currently no data to ascertain the long-term effect of drug treatments. Pharmaceutical treatments were coupled with some undesired side effects compared to placebo, but the side effects of psychological treatments are unknown. Direct comparisons between pharmaceutical and psychological treatments are needed as well as data to investigate the generalizability of these results to adolescents. Long-term follow-ups, standardized assessments including measures of quality of life, and the study of underrepresented populations should be a priority for future research.

Supplemental Information

Supplemental Information 1 Detailed information of the search strategy for the review.

Cochrane Library via Wiley 02 November 2015 (CDSR, DARE & CENTRAL). Updated search on November 2016. Title: Treatment of binge eating disorder.

Click here for additional data file.

Supplemental Information 2 List of all of the excluded studies.

Click here for additional data file.

Supplemental Information 3 Characteristics of the included studies in alphabetic order.

Details of the included studies in terms of population, inclusion criteria, setting, Intervention, duration, dropout, compliance, comparison conditions, duration of treatment, results, and risk of bias including comments on the study.

Click here for additional data file.

Supplemental Information 4 PRISMA checklist.

Click here for additional data file.

Additional Information and Declarations

Competing Interests

Author Contributions

Data Availability

The authors declare that they have no competing interests.

Ata Ghaderi conceived and designed the experiments, performed the experiments, contributed reagents/materials/analysis tools, prepared figures and/or tables, authored or reviewed drafts of the paper, approved the final draft.

Jenny Odeberg conceived and designed the experiments, performed the experiments, analyzed the data, contributed reagents/materials/analysis tools, prepared figures and/or tables, authored or reviewed drafts of the paper, approved the final draft.

Sanna Gustafsson performed the experiments, contributed reagents/materials/analysis tools, prepared figures and/or tables, authored or reviewed drafts of the paper, approved the final draft.

Maria Råstam performed the experiments, contributed reagents/materials/analysis tools, prepared figures and/or tables, authored or reviewed drafts of the paper, approved the final draft.

Agneta Brolund performed the experiments, contributed reagents/materials/analysis tools, authored or reviewed drafts of the paper, approved the final draft, literature search.

Agneta Pettersson performed the experiments, prepared figures and/or tables, authored or reviewed drafts of the paper, approved the final draft.

Thomas Parling performed the experiments, contributed reagents/materials/analysis tools, prepared figures and/or tables, authored or reviewed drafts of the paper, approved the final draft.

The following information was supplied regarding data availability:

All raw data used for the meta-analysis are provided in Appendix C.

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
