# Peer review of "Psychological, pharmacological, and combined treatments for binge eating disorder: a systematic review and meta-analysis"

_PeerJ, doi:10.7717/peerj.5113_

## Round 0.1 · original submission · Minor Revisions

Three reviews have now been received. All the reviewers agree that this systematic review with meta-analysis is timely and constitutes a contribution to the field. Reviewer 2 and 3 make specific points that they would like addressed, Reviewer 2 in their critique and Reviewer 3 in the annotations of the text. Please address the points raised by the reviewers and resubmit.

·

Basic reporting

The report is clear and concise. the authors have reported on all relevant material. The English is excellent. there are adequate references and much background and context are provided. Tables and Figures are clear.

Experimental design

The experimental design is sound. Tables and figures depict the experimental process.

Validity of the findings

I am not sure about the novelty of the results. My only real concern is that a previous meta-analysis was published only two years prior to this one. The authors claim that their meta-analysis and systematic review contains more relevant studies and is not focused only on RCTs. In this case it is worthy of publication.

Additional comments

This is a systematic review and meta analysis of psychological, pharmacological and combined treatments for BED. This study is important in the field of ED as current studies on these effects provide contradictory results.
This study is very interesting and should be published. The study is well written and provides important information concerning evidence based treatments. My only real concern is that a previous meta-analysis was published only two years prior to this one. The authors claim that their meta-analysis and systematic review contains more relevant studies and is not focused only on RCTs. In this case it is worthy of publication.

Reviewer 2 ·

Basic reporting

See general comments.

Experimental design

See general comments.

Validity of the findings

See general comments.

Additional comments

This was a systematic review and meta-analysis of psychological, pharmacological, and combined treatments for binge eating disorder. Most prior reviews have focused upon only one of these approaches to the treatment of BED, and as such, the present review is comprehensive in comparison. There are also several nice design features, including a solid search strategy, assessment of bias, and the grading of evidence. The conclusions concurred with most of the previous reviews on this topic.

1. The language was acceptable. Yet the manuscript was long, and could be revised to be more concise and economical in expression. The lines in the Discussion which refer back to the Introduction (522-540) could proabably be cut/merged with the similar info in the Introduction. The material in the Introduction from lines 64 to 75 could also be trimmed/cut—this was informative, but not immediately relevant to the aim. It is also a question of whether the Results could have been better and more succinctly summarized in a table instead of page after page of text. I defer to the authors and journal regarding this point.

2. Most of the implications were fairly research-oriented (i.e., need for longer follow-ups, trials should include adolescents, etc.), whereas concrete clinical implications were generally lacking. It would have been interesting to include some discussion regarding how findings match current treatment guidelines for BED such as NICE and other countries, e.g., Hilbert A, Hoek HW, Schmidt R. Evidence-based clinical guidelines for eating disorders: international comparison. Current Opinion in Psychiatry. 2017;30(6):423-437. doi:10.1097/YCO.0000000000000360.

3. There were many trials with acceptable risk of bias, but only 1 study or fewer than 100 participants, and these were not included in any of the meta-analyses. This was unfortunate, as it resulted in the exclusion of a sometimes high proportion of available studies (i.e., 13 of 24 psychological treatment trials), although some of these studies were promising. This conservative choice was acknowledged in the discussion, but perhaps another sentence or two at line 559 is warranted to expand upon the possible implications of the methodology applied.

4. A related point---the term “low certainty of the evidence” or “insufficient amount of studies” should be used (lines 230 to 270) instead of “insufficient scientific evidence”. The latter term may be interpreted as having non-significant results----and cause some confusion.

5.Table 1. I would recommend to explain all abbreviations (e.g., CBT, SMD, RD, BE, etc) in the footnote so the table stands alone, without having to refer to the text.

6. In addition to your explanation of risk difference, I would also recommend explaining (on lines 271-276) the benchmarks for interpreting SMD, as interpretation was a bit difficult in the absence of a funnel plot.

·

Basic reporting

This is a well articulated article, a pleasure to read. It explores an important area of clinical significance, looking at the psychological, pharmacological, and combined treatments for binge eating disorder. Given the prevalence of binge eating disorders in obesity. this review is timely and worthwhile. The abstract was clear and concise and provided a comprehensive summary of entire study. Overall, the introduction was well constructed. All salient constructs well described and there was seamless integration of concepts. The rationale and justification for the present review was made clear. The introduction concluded with a clearly stated aim appropriate to the research methodology. However Fairburn's CBT E model needs a mention in psychological interventions. Some minor edits have been suggested in language and referencing style (in text comments).Line 123 and 119 seem to contradict each other. Lines 126-132 seem out of place and perhaps belong in the Discussion section?

Experimental design

The method was clearly written and contained all relevant information appropriate to the methodology for a systematic review and MA. Relevant information pertaining to study design, eligibility criteria, data collection and data analysis were all clearly described. Again some in-text comments have been provided for minor edits.

Validity of the findings

Generally the results were very well presented. There was a very good summary of the limitations of the study.

Additional comments

This review has demonstrated a good level of analysis as well as potential applications back to the real-world and clinical settings in its comparison of psychological, pharmacological, and combined
treatments for binge eating disorder.Well done ! Some minor edits suggested in in-text comments.

---

## Round 0.2 · accepted · Accept

Thank you very much for submitting this paper to PeerJ and for making the revisions requested by the reviewers. I think this will make a valuable contribution to the understanding and treatment of BED.

#